# Refining of Precious Metal Bearing Materials from Secondary Sources-Methanesulfonic Acid Leaching of Raw Silver Granules as a Promising Approach towards a Green Way of Silver Refining

**DOI:** 10.3390/ma14206095

**Published:** 2021-10-15

**Authors:** Johannes Hopf, Aaron Weigelt, Hartmut Bombach, Michael Stelter, Alexandros Charitos

**Affiliations:** Institute for Nonferrous Metallurgy and Purest Materials, TU Bergakademie Freiberg, Leipziger Straße 34, 09599 Freiberg, Germany; Aaron.Weigelt@inemet.tu-freiberg.de (A.W.); h.bombach@web.de (H.B.); Michael.Stelter@inemet.tu-freiberg.de (M.S.); Alexandros.Charitos@inemet.tu-freiberg.de (A.C.)

**Keywords:** hydrometallurgy, raw silver refining, leaching, methanesulfonic acid, hydrogen peroxide, green metallurgy

## Abstract

The state-of-the-art technology of raw silver refining in a silver nitrate-based electrorefining process (Moebius-electrolysis) is accompanied by several disadvantages, both from a technological and from an ecological point of view. In addition, increasing concentrations of critical impurities from secondary sources, like palladium, in raw silver are a further challenge for the future of silver refining. Thus, there is strong motivation for the development of an adequate, alternative process of raw silver refining to substitute the existing Moebius-electrolysis. Due to its less environmentally toxic character and the high aqueous solubility of its silver salt, methanesulfonic acid (MSA) is a possible base chemical for the design of an efficient refining method based on leaching of raw silver followed by electrowinning, with less ecological and technological complications. In this paper the results of some fundamental investigations on the leaching of raw silver granules, containing approx. 94% silver, with methanesulfonic acid and hydrogen peroxide as an oxidation agent are presented. Agitation leaching experiments were conducted on a laboratory scale and the effects of the solid concentration, the hydrogen peroxide dosage and the temperature as leaching parameters were studied. The obtained results indicate that silver leaching yields of more than 90% are achievable with leaching at elevated temperatures of 65 °C or 80 °C, solid concentrations of 500 g/L and at a stoichiometric H_2_O_2_:Ag-ratio of 3:1. Increased solid concentrations greater than 500 g/L and elevated temperatures of 65 °C or 80 °C additionally improved the selectivity of the process regarding the leaching of Pd.

## 1. Introduction

Although silver is an element that plays an indispensable role in the function of modern industrial economies and societies with an annual production of approx. 25,000 t in 2020 [1] and a wide field of applications, e.g., as a conductor in electronics industries or as a component of several catalysts [2], few investigations have been made to develop an alternative refining technology to overcome the existing disadvantages of the Moebius-electrolysis, that are mainly the cathodic growth of silver dendrites, the handling of anode bags and scrapers, the re-dissolution of deposited silver by nitric acid and the treatment of evolving hazardous nitrous gases and nitrate bearing waste waters [3]. A possible alternative technology for the refining of raw silver could consist of the leaching of raw silver and a subsequent electrowinning process of the silver-rich leaching liquor to obtain fine silver. Efforts that have been made with the scope of developing an alternative silver refining technology preliminarily showed that methanesulfonic acid could be a possible substitute for nitric acid. For example, Dressler et al. showed that silver electrolysis in a methanesulfonic acid-based silver electrolyte resulted in an improved quality of the silver cathodes and a higher current efficiency than in the Moebius-electrolysis [4]. In addition to the technological advantages coming along with the use of MSA, the ecologically beneficial aspects like its biodegradability and the absence of the evolution of hazardous gases make it an even more desirable candidate for silver refining in terms of a green industry [5].

These promising findings give an impetus for further investigations on the refining of raw silver in a MSA-based system based on a leaching step and electrowinning to replace the traditional Moebius-electrolyis. Thus, the goal of this paper is to show the suitability of MSA for leaching of raw silver as a part of the alternative refining technology. Fundamental leaching parameters like solid concentration, temperature and hydrogen peroxide dosage are investigated. Since recycling material from secondary sources shows increasing concentrations of Pd, which is a crucial impurity in silver refining due to its similar electrochemical behaviour in comparison to Ag and thus is a serious challenge for the metallurgical industry [6], special attention is paid to the selectivity of the investigated leaching process regarding the leaching of Pd.

## 2. Methanesulfonic Acid in Leaching and Metal Recovery—A Literature Review

MSA is a strong semi-organic acid with a pK_A_ ≈ −1.9 [7]. Many of its metal salts show high solubility in water. For example, the maximum aqueous solubility of silver-methanesulfonate at 22 °C is 713 g/L, which corresponds to about 400 g/L Ag^+^ [5]. Furthermore, it provides advantageous physical and chemical properties like high thermal stability, low vapor pressure and strong resistance to oxidizing agents like hydrogen peroxide [7]. Among others, these properties established the use of MSA in various hydrometallurgical and electrometallurgical processes. Several authors reported the use of MSA in the leaching of malachite [8], chalcopyrite [9], jarosite [10], galena [11], cerussite [12], hemimorphite [13] or in the refining of lead [14].

From the fact that MSA is an acid without oxidizing character arises the necessity of the addition of an oxidizing agent for leaching of precious metals like silver. Due to its high oxidation potential, hydrogen peroxide is an appropriate oxidizing agent for the oxidation of silver [15]. The only reaction product of its consumption is water, which additionally contributes to the environmentally beneficial aspects of the investigated leaching process. The equations for the corresponding reactions of oxidation, reduction, and the overall reaction are given below:(1)$$\mathrm{Ag}\to {\mathrm{Ag}}^{+}+e^{-}\;\left(E^{0}=0.799\;V\right)$$
(2)$$H_{2}O_{2}+2{\;H}^{+}+2{\;e}^{-}\to 2{\;H}_{2}O\;\left(E^{0}=1.763\;V\right)$$
(3)$$2\;\mathrm{Ag}+H_{2}O_{2}+2{\;H}^{+}+2{\;\mathrm{SO}}_{3}{\mathrm{CH}}_{3}{}^{-}\to 2{\;\mathrm{Ag}}^{+}+2{\;\mathrm{SO}}_{3}{\mathrm{CH}}_{3}{}^{-}+2{\;H}_{2}O\;$$

In the literature it was reported that different silver-containing materials were successfully leached with MSA and H_2_O_2_ according to Equation (3). Silver, from spent Si-solar cells containing 0.08% Ag and major amounts of Si, Al, and Cu, was leached with MSA and H_2_O_2_ at different MSA:H_2_O_2_-ratios by Yang et al. at 25 °C with a 1 h to 12 h leaching time. They claimed the MSA:H_2_O_2_-ratio to be the most important parameter for the leaching and obtained optimum Ag-extraction yields with a 90:10-ratio of MSA (99%):H_2_O_2_ (30%) [16].

Schosseler et al. successfully used MSA and H_2_O_2_ to leach silver from preliminary processed oxygen depolarized cathodes containing 91% Ag, 0.3% Ni and 5.4% PTFE. The authors focused on the influence of parameters like temperature, the amount of hydrogen peroxide, and the leaching time on the leaching yield and maintained a solid concentration of 50 g/L. For a temperature of 35 °C, a 3-times stoichiometric H_2_O_2_-surplus and an 8 h leaching time, they obtained Ag leaching yields up to 95% and found the temperature and the stoichiometric ratio of H_2_O_2_:Ag to have significant impact on the leaching yield; with a lower H_2_O_2_:Ag ratio and with rising temperatures, the yield decreased due to the accelerated decomposition of H_2_O_2_ at higher temperatures [17].

An undesired side reaction occurring during leaching is the exothermic decomposition of H_2_O_2_ on the surface of metallic particles like the raw silver granules and particles of the leaching residue [15]:(4)$$2{\;H}_{2}O_{2}\to \;2{\;H}_{2}O+{\;O}_{2}\;\Delta H=-196.2\frac{\mathrm{kJ}}{\mathrm{mol}}$$

Consequently, there is always a certain amount of added H_2_O_2_ that does not contribute to the intended leaching process.

## 3. Calculation of the Leaching Yield

One way to calculate the leaching yield $\eta_{X}$ of an element X is by dividing the leached mass of the element $m_{X}$ (based on the concentration of the element in the leaching solution at the end of the leaching process $c_{X}$ and the volume V of the solution) by the initial mass of this element in the raw silver granules $m_{X0}$:(5)$$\eta_{X}=\frac{m_{X}}{m_{X0}}\cdot 100\%=\frac{c_{X}\cdot V}{m_{X0}}\cdot 100\%$$

Another way to determine the leaching yield is the calculation based on the mass of the element left in the solid leaching residue after the leaching process $m_{\mathrm{XE}}$:(6)$$\eta_{X}=\frac{m_{X0}-m_{\mathrm{XE}}}{m_{X0}}\cdot 100\%$$

In addition, the ratio between the total mass of the leaching residue $m_{\mathrm{residue}}$ and the initial mass of the raw silver granules $m_{\mathrm{granules}}$ can be considered a key indicator for the overall success of the leaching process:(7)$$\eta_{\mathrm{overall}}=\frac{m_{\mathrm{residue}}}{m_{\mathrm{granules}}}\cdot 100\%$$

There is a theoretical minimum value of about 3.51 wt.-% for this ratio corresponding to the fraction of gold in the raw silver (see Section 4), since gold is the only completely insoluble element in the raw silver under the given thermodynamic leaching conditions [15].

## 4. Materials and Method

Raw silver granules produced from the same raw silver bulk from which silver anodes are usually cast for the Moebius-electrolysis were used for the leaching experiments. Their particle size was < 1 mm (100%) and their chemical composition—as provided by the project partner—is given in Table 1.

The other base chemicals used for the leaching experiments were MSA (70%) and H_2_O_2_ (50%), both provided by VWR Chemicals (Darmstadt, Germany) (CAS: 75-75-2 and 7722-84-1, respectively).

All leaching experiments were carried out in a 1 L double walled glass vessel with an overhead electric stirring unit and a PTFE stirrer. The stirring speed for all experiments was 650 rpm.

Initially, the vessel was filled with the desired amounts of raw silver granules and diluted MSA, with the desired acid concentration. The amount of MSA used for leaching was adjusted in such a way that the concentration was 100 g/L above the necessary concentration for the theoretical dissolution of the whole silver (according to Equation (1)), i.e., the total acid concentration varied with the solid concentration. For example, for 50 g of raw silver and 0.1 L of acidic solution, this corresponded to a total MSA-concentration of 545.5 g/L. The desired amount of H_2_O_2_ (50%) was continuously pumped by a peristaltic pump into the vessel with a pumping rate of approx. 0.5 mL/min. For the leaching experiments at a constant temperature, the latter was maintained by temperature-controlled water in the double walled glass vessel, continuously circulating between the reactor and a connected thermostat. For the experiments varying the solid concentration, there was no temperature control, i.e., no thermostat was used. The temperature inside the vessel and inside the thermostat were measured with NiCr/Ni-thermocouples, respectively, and data recorded with a data logger (Ahlborn Almemo^®^ 2590A, Ahlborn Mess-und Regelungstechnik GmBH, Holzkirchen, Germany). The leaching time was 5 h for each experiment, if not stated otherwise. The technical equipment and the setting for the leaching experiments are shown in Figure 1.

Several samples of the leaching solution were taken at different points in time throughout each experiment. Ag concentrations in the samples of the leaching solutions were determined by potentiometric titration with 0.05 N HCl; concentrations of Pd, Pt, Pb and Cu were determined by inductively coupled plasma optic emission spectroscopy (ICP-OES, Varian 725-ES, Agilent, Santa Clara, CA, USA). The leaching residue was washed with deionized water, dried, weighed, and then analyzed by digestion in aqua regia with measurement of the solution via ICP-OES.

The investigated parameters in this paper were the solid concentration, the hydrogen peroxide dosage, and the temperature. Leaching yields for several elements were calculated and are discussed in Section 5.

## 5. Results and Discussion

### 5.1. The Effect of the Solid Concentration

The obtained leaching yields and the ratio of the mass of the leaching residue and the raw silver granules for different values of the solid concentration are shown in Figure 2.

The leaching yields and the relative masses of the leaching residues were similar for the different values of the solid concentration from 100 g/L to 550 g/L. Silver leaching yields of about 90% and higher were obtained, with a maximum leaching yield of about 95% for 550 g/L. The ratio of the mass of the residue and the mass of the raw silver granules was in the range between 8 and 9% for the named solid concentrations. At a solid concentration of 600 g/L, the leaching deteriorated, indicated by a decrease in the silver leaching yield to 82% and an increase of the relative mass of the leaching residue to about 16%. An explanation for this effect could possibly be the limited solubility and oversaturation of the silver salts in the acidic solution due to the greater amount of processed raw silver. In addition, the solubility of the silver salts depends on the total acid concentration [5]; as the acid concentration is higher at higher solid concentrations according to the experimental setup, it also has an influence on the leaching performance. Furthermore, the intensive contact between solid and liquid phase induced by the stirrer is limited up to a certain value of the solid concentration.

Another finding is the improvement of the selectivity of the leaching process at higher solid concentrations regarding Pd. Pd is co-leached along with the silver because it is also oxidized by hydrogen peroxide [15]:(8)$$\mathrm{Pd}\to {\mathrm{Pd}}^{2+}+2e^{-}\;\left(E^{0}=0.915\;V\right)$$

The leaching yield of Pd is 21% at a solid concentration of 100 g/L, it decreases to 7% at 550 g/L, whereas the silver leaching yield remains at values higher than 90%. Hence, leaching at high solid concentrations, up to 550 g/L, on one hand allows improving the space-time-yield of the whole leaching process while maintaining satisfying leaching results, and on the other hand has an advantageous effect on the separation of Pd from the silver-bearing leaching liquor, which is later treated by an electrowinning or cementation process, where Pd is an unwelcome and critical impurity due to the danger of its co-deposition and the possible contamination of the cathodic fine silver.

### 5.2. The Effect of the H_2_O_2_ Dosage

The results from leaching at different H_2_O_2_:Ag-ratios can be seen in Table 2. Except the last case, where H_2_O_2_ was added at three definite points of time (0 h, 1 h, and 3 h, respectively), H_2_O_2_ was added continuously in all leaching experiments.

Leaching with smaller amounts of H_2_O_2_ resulted in lower Ag leaching yields and thus higher masses of the leaching residue, as a portion of the H_2_O_2_ always decomposes catalytically, according to Equation (4). The best leaching results were obtained with a 3-times stoichiometric H_2_O_2_:Ag-ratio, which is in accordance with the results of Schosseler et al., who claimed a 3:1 ratio to be optimal [17].

When H_2_O_2_ is added in bigger portions at several points of time instead of by continuous pumping, the overall leaching performance deteriorates significantly because of the strong sudden heat generation due to the exothermic, catalytic decomposition according to Equation (4). Since the stability of H_2_O_2_ decreases with higher temperatures, the exothermic decomposition auto-accelerates itself [15]. In contrast, Schosseler et al. found that adding the whole amount of H_2_O_2_ at a few points in time during the experiment could also result in high silver leaching yields of up to 95% [17].

### 5.3. Temperature Development during Leaching at Constant Temperatures

In Figure 3 a typical curve for the development of the temperature inside the reaction vessel during leaching at a constant temperature is shown. The exothermic reaction of the H_2_O_2_ pumped into the vessel causes heat generation that leads to a sharp increase of temperature within the first 15 min of the experiment. The oxidation of silver is initiated when the accumulated amount of H_2_O_2_ is sufficient to reach the needed oxidation potential in the acidic solution. This results in the sharp peak at the beginning of the curve. Later, the temperature decreases to an almost constant level, as long as H_2_O_2_ is added and the generation of heat (both from the reaction and the catalytic decomposition of H_2_O_2_, according to Equation (2) and Equation (4), respectively) is in progress. With the end of the H_2_O_2_-dosage after approx. 1.5 h, the temperature decreases to its original level. The gap between the actual temperature inside the vessel and the intended temperature of the water from the thermostat is due to heat exchange between the temperature-controlled water on its way to the vessel and the environment.

In Figure 4 the corresponding curves for leaching experiments at different constant temperatures are compared. The chosen temperatures were 11 °C, 25 °C, 45 °C, 65 °C and 80 °C. One curve shows the temperature for leaching without temperature maintenance by a thermostat.

Apparently, the observed temperature peak occurring at the beginning of the leaching experiments is present at all investigated leaching temperatures. At higher temperatures, the height of the peak, i.e., the temperature increase due to the reaction of the added H_2_O_2_, is smaller due to the already higher temperature level of the surrounding leaching solution. Another observation is that at higher temperatures, the temperature peak occurs earlier. This, in conclusion, suggests—as the dosage rate of H_2_O_2_ is the same in all cases—that the necessary oxidation potential to initiate the reaction is reached earlier at higher temperatures because of the improved reactivity of the leaching reactants. Again, all the curves show a temperature drop to the respective initial temperature level with the end of the H_2_O_2_-dosage after approx. 1.5 h.

The temperature curve for the leaching experiment without temperature control expectedly deviates: The temperature also increases sharply when the necessary oxidation potential is reached, but then stays at a higher level (approx. 54 °C) until the end of the H_2_O_2_-dosage before the solution cools down to room temperature at the end of the leaching experiment.

### 5.4. Leaching Results for Leaching at Different Temperatures

In Table 3, the ratio between the mass of the leaching residue (after leaching) and the mass of the raw silver granules (before leaching) is shown for each experiment, respectively. The mass of the leaching residue decreases with increasing leaching temperature, which is an indicator for the improvement of the total leaching yield at higher temperatures due to the higher reactivity of the components. The ratio decreases from almost 60% at 11 °C to approx. 9% at 80 °C. These findings indicate that the intensification of the reactivity seems to overcompensate the acceleration of the decomposition of H_2_O_2_ at elevated temperatures. These results are in contrast to the findings of Schosseler et al., who claimed the silver leaching yield decreased by 19% as the leaching temperature increased from 35 °C to 50 °C [17]. Previous investigations on the leaching of the raw silver granules found that under optimum leaching conditions, there is a range between 8% and 10% for the above named mass ratio, indicating a maximum leaching yield that cannot be exceeded with the chosen materials and conditions.

Discussing the results in more detail: in Figure 5 and Figure 6 the leaching yields of several elements contained in the raw silver are shown and in one case determined by analysis of the corresponding leaching residue via digestion in aqua regia and ICP-OES-analysis of the digestion solution (Figure 5), and in the other case determined by analysis of the leaching solution itself, also via ICP-OES (Figure 6). The results based on the analysis of the leaching residue deceptively suggest that gold was also leached from the raw silver. However, thermodynamically the dissolution of gold is not possible under the given conditions. No dissolved gold was found in any leaching solution. The errors probably originate from an incomplete digestion of the leaching residue during aqua regia digestion and consequently from a distortion of the mass balance. Unleached granules, that were mainly present in the leaching residue after leaching at lower temperatures, complicate the aqua regia digestion due to entrapped gold in silver, which cannot be attacked by the aqua regia since silver forms an insoluble surface layer of AgCl. Nevertheless, although there seems to be a systematic error, an apparent tendency of decreasing leaching yields of the platinum group metals (Pd and Pt) with increasing temperatures can be seen in the diagram.

For the obvious inaccuracy of the mentioned analysis method, the leaching yields based on the analysis of the leaching solutions were calculated and are shown in Figure 6. Compared to the leaching yields calculated based on the solid residue, there is a visible difference for all investigated elements due to the reasons mentioned before. The leaching yield of silver increases up to 90% and more with increasing temperatures. Again, a tendency of decreasing leaching yields of platinum group metals at higher temperatures can be seen when calculating the leaching yield by analysis of the concentration of elements in the final leaching solution.

The depletion of Pd in the leaching solution over time at high leaching temperatures can also be seen in Figure 7. After a first increase, the concentration reaches a peak and then decreases again until the end of the respective leaching experiments, while the overall leaching yield stays in an acceptable range, as indicated by the ratio between leaching residue and raw silver (see Table 3). The reasons for the observed behavior of Pd and Pt are a current object of investigations and are not discussed here. However, these findings allow choosing leaching parameters that ensure a high leaching yield of silver and, at the same time, improve the selectivity of the leaching process regarding the separation of silver and palladium.

In Table 4, the chemical composition of the leaching residue from an experiment with optimum leaching results are shown. During leaching, metals like Au, Pt and Pd enrich by a factor higher than 10 in the leaching residue, whereas silver, copper, and lead are mainly oxidized and dissolve in the leaching solution.

A silver leaching yield of 100% was not reached even under the most effective leaching conditions. The reason for this may be the partial entrapment of silver in gold, which does not dissolve during leaching. Additionally, aiming for higher silver leaching yields always comes along with the risk of the simultaneous leaching of more undesired elements, like Pd, impairing the selectivity of the process. For the process design, the rest of the silver in the leaching residue is not a critical issue, since the residue itself will also be processed in further steps to extract and separate the contained noble metals.

## 6. Conclusions and Outlook

Raw silver granules, containing approx. 94% of silver along with other noble metals like Au and PGMs, were successfully leached with methanesulfonic acid and hydrogen peroxide as an oxidation agent. Leaching yields higher than 90% were reached with solid concentrations up to 550 g/L and a 3-times stoichiometric H_2_O_2_ surplus. At higher solid concentrations the selectivity of the leaching process regarding Pd increased. Additionally, leaching at several temperatures from 11 °C to 80 °C was investigated. The overall leaching yield increased with higher temperatures and reached its optimum in the range between 65 °C and 80 °C, leaving a residue with a mass of less than 10% of the raw silver. Silver leaching yields of more than 90% were reached under those conditions. At higher temperatures, again, a beneficial increase of the selectivity of the leaching process regarding the leaching of Pd was observed. The results show that methanesulfonic acid may be an appropriate alternative base chemical for developing an ecologically less hazardous technology of silver refining, and may even be suitable for materials from secondary sources with higher concentrations of impurities like Pd. Further investigations are necessary to clarify the observed mechanisms of the depletion of Pd during leaching, to develop the following step of silver extraction from the leaching solution, and to study the effect of impurities like Cu and Pd on leaching performance when recycling the electrolyte.

## Figures and Tables

**Figure 1 materials-14-06095-f001:**
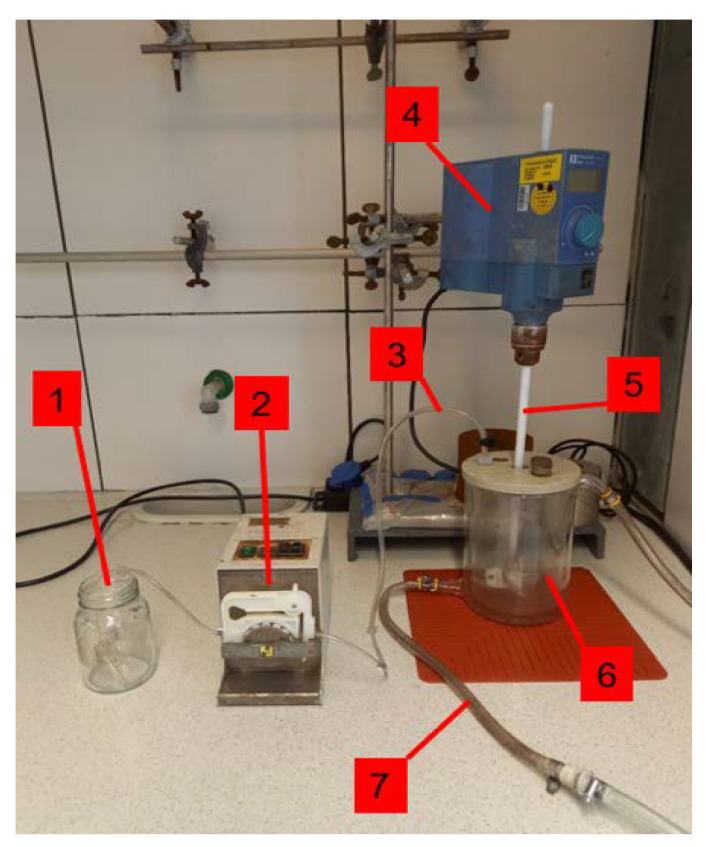
Leaching equipment. 1—H_2_O_2_ reserve, 2—peristaltic pump, 3—tubes and inlet for H_2_O_2_, 4—stirrer engine, 5—PTFE-stirrer, 6—glass vessel (V: 1 L), 7—tubes to the thermostat.

**Figure 2 materials-14-06095-f002:**
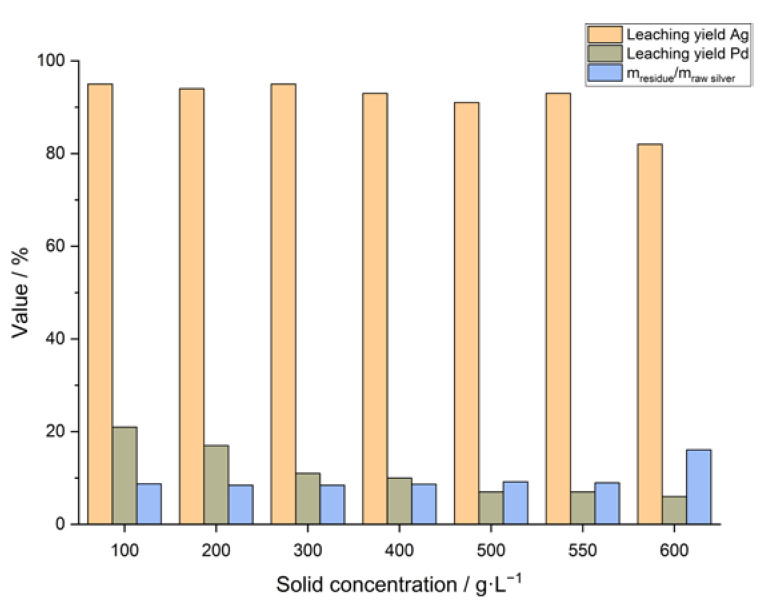
Summary of the leaching results for different solid concentrations.

**Figure 3 materials-14-06095-f003:**
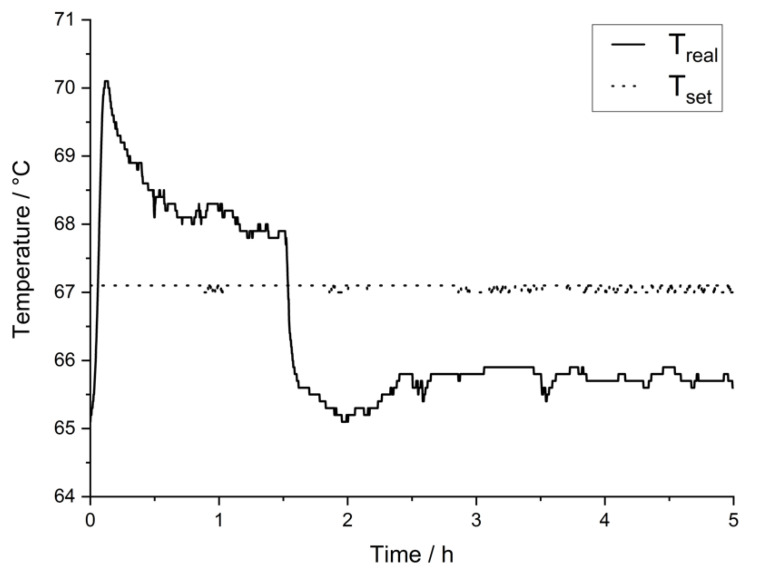
Detailed temperature curve for leaching at T: 65 °C; solid conc.: 50 g/0.1 L.

**Figure 4 materials-14-06095-f004:**
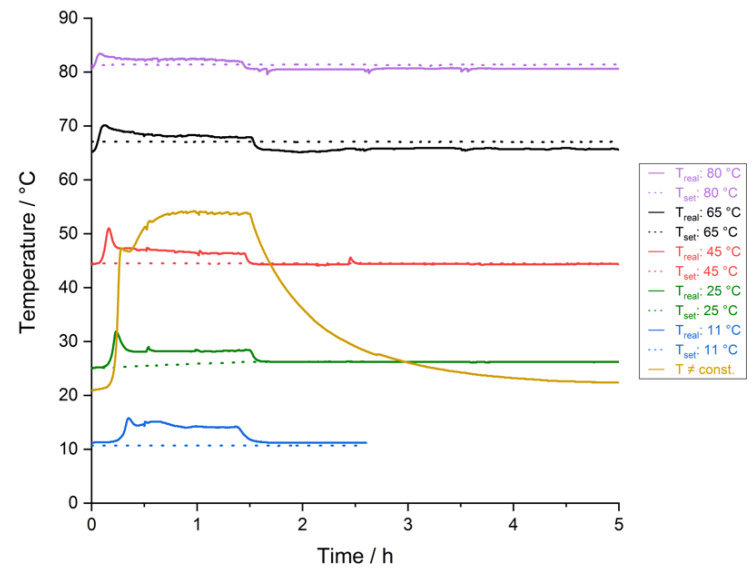
Temperature curves for leaching at different temperatures (dotted lines: thermostat; thick lines: reactor); solid conc.: 50 g/0.1 L.

**Figure 5 materials-14-06095-f005:**
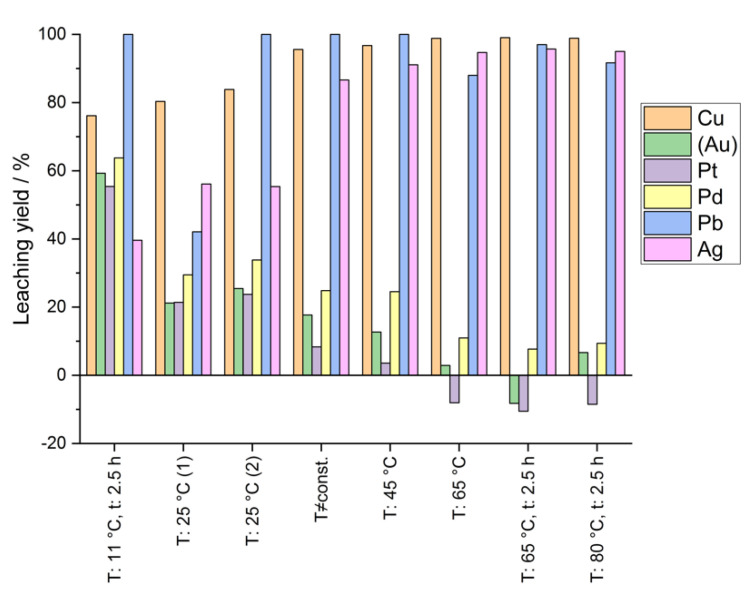
Leaching yields of several elements at different leaching temperatures (calculation based on analysis of leaching residue via aqua regia digestion); solid conc.: 50 g/0.1 L; t: 5 h.

**Figure 6 materials-14-06095-f006:**
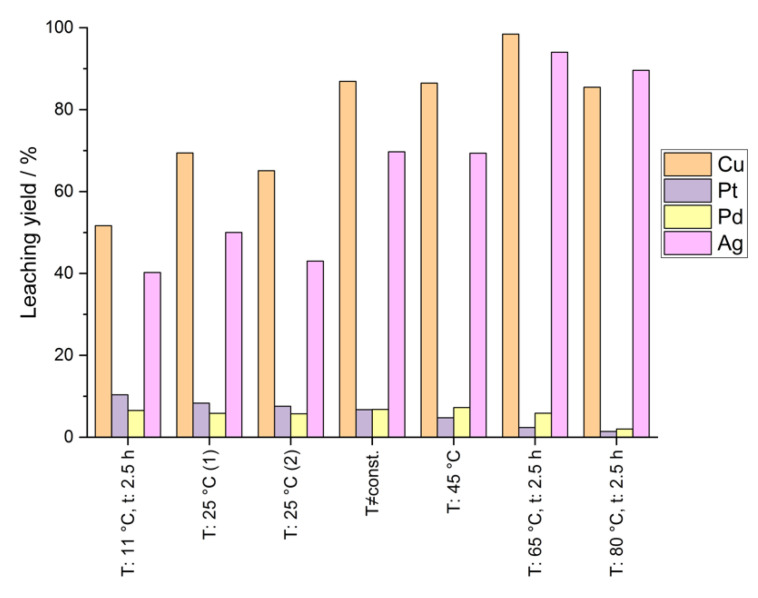
Leaching yields of several elements at different leaching temperatures (calculation based on analysis of liquid samples); solid conc.: 50 g/0.1 L; t: 5 h.

**Figure 7 materials-14-06095-f007:**
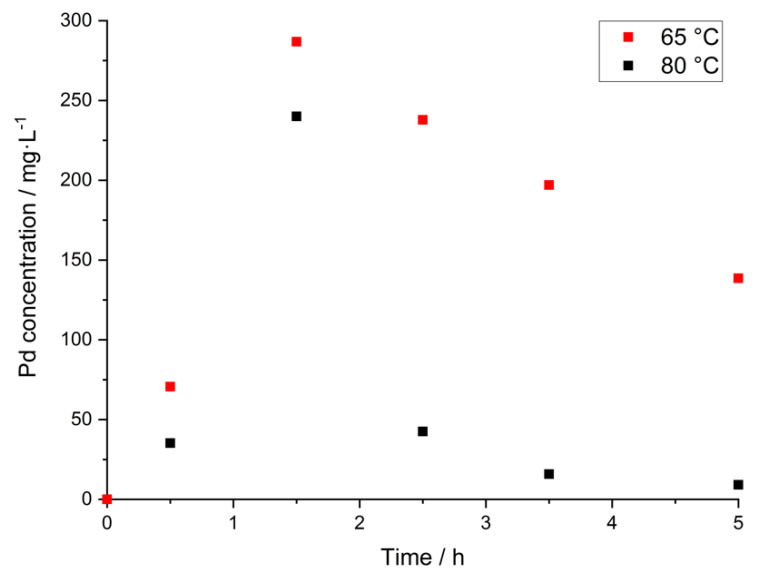
Pd-concentration during leaching at higher temperatures; solid conc.: 50 g/0.1 L; H_2_O_2_:Ag: 3:1; c _MSA_: 100 g/L above the stoichiometric amount.

**Table 1 materials-14-06095-t001:** Chemical composition of the raw silver granules.

	Ag	Au	Pd	Pt	Cu	Pb	∑
wt.-%	93.86	3.51	0.93	0.18	1.41	0.02	99.91

**Table 2 materials-14-06095-t002:** Leaching results for different values of the stoichiometry. H_2_O_2_/Ag-ratio (Leaching time = 5 h, solid conc. = 300 g/L).

H_2_O_2_/Ag Stoich. Ratio (-)	m_residue_/m_raw silver_ (%)	Leaching Yield Ag (%)	Leaching Yield Pd (%)
3:1	8.43	95	11
2.5:1	9.93	92	11
1:1	18.90	83	9
3:1 (*)	42.84	59	6

* H_2_O_2_ added three times.

**Table 3 materials-14-06095-t003:** Mass ratio of leaching residues and raw silver granules for leaching at different temperatures; solid conc.: 50 g/0.1 L.

Experiment	Temperature (°C)	m_residue_/m_raw silver_ (%)
1	11 (t: 2.5 h)	58.88
2	25	45.06
3	25	45.50
4	T ≠ const.	16.36
5	45	12.36
6	65	9.42
7	65 (t: 2.5 h)	8.90
8	80 (t: 2.5 h)	9.02
9	80	9.28

**Table 4 materials-14-06095-t004:** Chemical composition of a typical leaching residue (optimum leaching conditions).

	Ag (Balance)	Au	Pd	Pt	Cu	Pb	∑
wt.-%	45.28	42.68	9.688	2.186	0.155	0.007	100

## Data Availability

The data presented in this study are available on request from the corresponding author.

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
