# Peer review of "Refining of Precious Metal Bearing Materials from Secondary Sources-Methanesulfonic Acid Leaching of Raw Silver Granules as a Promising Approach towards a Green Way of Silver Refining"

_materials, 2021, doi:10.3390/ma14206095_

Round 1

Reviewer 1 Report

The article proposed by the authors is interesting and original. The results are fairly well presented and the discussion of them is adequate. In fact, I only have two comments in relation to this manuscript: 1) there is no statistical processing of the data. Have the tests been carried out in triplicate? if so, the error bars should be shown in the figures; 2) the authors insist on the selective leaching of silver relative to palladium. However, significant solubilizations of copper and lead were observed along with silver. Will this be a problem? how could this problem be solved?

Author Response

Hello! Regarding your two comments:
1) I did not do statistical processing of the data, so there are no error bars.
2) The dissolution of copper and lead along with silver under the given leaching conditions is unavoidable, since both elements are less noble than silver and the solubility of their MSA-salts is high. Nevertheless, there is no problem with this co-dissolution: These elements don't cause trouble in the following electrowinning-process, since their electrochemical standard potential is much lower than the one of silver and they do not deposit on the cathode. The difference to Pd in this case is its similar standard potential compared to Ag, which causes the problem in electrowinning. However, if Cu and Pb enrich in the electrolyte during the circulation between several leaching and electrowinning steps, a crucial concentration can be reached that enhances the co-deposition (especially in the case of Cu). Before reaching this concentration the electrolyte should be purified to get rid of these elements, e.g. by cementation of the Cu.

Best regards

Reviewer 2 Report

I was asked to review the following manuscript. hHere're some comments:

  1. State-of-art should be improved as MSA is relativly "hot topic" for metal recovery and in context to precious metals it was broadly examined i.a. by Binnemanns, Friedrich, Cho and others.
  2. Authors should pint out what is the main goal of this manuscript and why is it better to other work in literature.
  3.  Paragraph 3. calculation -- these are simple equations and need nothing more than 3-5 lines rather than 1/2 page.
  4. Paragraph 5.1 please indicate why it was not possible to achieve total leaching (i.e. Efficiency 100%).
  5. Line 175"the capacity f PTFE stirrer to ensure the motion" - do not tryy to complicate sentence too much. Make it simple.
  6. What is the reason of Pd co-leaching and why it is only 6-11%?
  7. Somewhere in the text, probably in conclusions there's "...leachin at several temperatures from 11 to 80C...". I guess it was thermostatted but 11 (not 10 or 15)

Author Response

1. State-of-art should be improved as MSA is relatively "hot topic" for metal recovery and in context to precious metals it was broadly examined i.a. by Binnemanns, Friedrich, Cho and others.

The literature we have focussed on for the review of the state-of-the-art is all dedicated to research on leaching of silver with MSA and a necessary oxidation agent. We edited the text and added a brief review concerning the use of MSA in leaching of other materials, as has been proposed by the reviewer. Selected works of Binnemanns, Dreisinger and others have been references. Especially, the work of Friedrich (main author Schosseler) has been referenced several times within the text. The above actions have led to an increase of the reference list by 7 references. We feel that a more extensive review on MSA leaching of feed materials other than the silver granules would be beyond of the scope of the review. Nonetheless, the brief review included as suggested by the reviewer gives a fuller picture with regard to MSA application.

2. Authors should point out what is the main goal of this manuscript and why is it better to other work in literature.

We have taken the reviewer comment into account. Hence, we added a few sentences in the abstract and in the introduction to point out the goal (substitution of Moebius-electrolysis by alternative raw silver refining technology consisting of leaching and electrowinning in MSA) more precisely.

This novel processing route for silver granules, part of which is leaching with MSA has been highlighted within the text in lines 17-19, 40-43, 54-55.

3. Paragraph 3. calculation -- these are simple equations and need nothing more than 3-5 lines rather than 1/2 page.

We agree with the reviewer that the representation of the equations could be more compact. We have edited the paragraph, thus producing a condensed summary. Nevertheless (as it is important for the rest of the discussion) we want to maintain the emphasis on the difference between the two ways of calculating the leaching yield. 

4.Paragraph 5.1 please indicate why it was not possible to achieve total leaching (i.e. Efficiency 100%).

The reviewer raises a valid point with regard to the leaching process. We commented on this in the lines 318-320: “A silver leaching yield of 100 % was not reached even at the most effective leaching conditions. The reason for this may be the partial entrapment of silver in gold, which does not dissolve during leaching.”

5. Line 175"the capacity f PTFE stirrer to ensure the motion" - do not try to complicate sentence too much. Make it simple!

We have edited and simplified the sentence in line 182 to read “Furthermore, the intensive contact between solid and liquid phase induced by the stirrer is limited up to a certain value of the solid concentration.”.

6. What is the reason of Pd co-leaching and why it is only 6-11%?

The reason for the co-leaching of Pd is the similar electrochemical standard potential of Pd compared to Ag (Ag/Ag+ E°= 0.799 V; Pd/Pd2+ E°=0.915). It is oxidized by hydrogen peroxide, too (we edited the text and added the information in line 187). Since there is a great excess of Ag in the raw silver, Ag is preferably leached. The co-leaching of Pd is unavoidable, but the general lesser leaching yield of Pd compared to Ag is due to the excess of Ag and the more noble character of Pd.

7. Somewhere in the text, probably in conclusions there's "...leaching at several temperatures from 11 to 80C...". I guess it was thermostatted but 11 (not 10 or 15)

We understand the comment of the reviewer. However, it should be noted that in fact the process temperature was 11 °C. The cryostat was set on the lowest possible temperature, but due to the higher room temperature (> 20 °C) the cooling water got a bit warmer on its way to the double wall glass vessel, so that the temperature of the leaching solution was 11 °C. We edited the text appropriately to clarify this aspect (line 225-228): “The gap between the actual temperature inside the vessel and the aimed temperature of the water from the thermostat is due to heat exchange between the temperature-controlled water on its way to the vessel and the environment.”.
